# Canonical WNT signalling governs *Echinococcus* metacestode development

**Ruth Herrmann[1], Michaela Herz[1], Kilian Rudolf[1], Akito Koike[1], Markus Spiliotis[1], Monika Bergmann[1], Nancy Holroyd[2], Uriel Koziol[1,3], Matt Berriman[2¤], Klaus Brehm[1]\***

**1** Consultant Laboratory for Echinococcosis, Institute of Hygiene and Microbiology, University of Würzburg, Würzburg, Germany, **2** Parasite Genomics, Wellcome Sanger Institute, Wellcome Genome Campus, Hinxton, Cambridge, United Kingdom, **3** Sección Biología Celular, Facultad de Ciencias, Universidad de la República, Montevideo, Uruguay

¤ Current address: School of Infection & Immunity, College of Medical, Veterinary and Life Sciences, University of Glasgow, Glasgow, UK

\* klaus.brehm@uni-wuerzburg.de

## Abstract

Alveolar echinococcosis (AE) is a lethal zoonosis caused by infiltrative growth of the metacestode larva of the tapeworm *Echinococcus multilocularis* in host organs. We previously showed that the *Echinococcus* metacestode is an evolutionarily unique, broadly posteriorized tissue, leading us to hypothesize that canonical WNT (cWNT) signalling, which patterns the body axis across metazoans, might be critical for metacestode formation. Here, we report effective RNAi-mediated knockdown of the *E. multilocularis* β-catenin gene (*bcat-1*), the central effector of cWNT signalling, in a primary parasite cell culture system that produces metacestode vesicles. *bcat-1*(RNAi) cultures were markedly impaired in vesicle formation, exhibited stem-cell hyperproliferation, and displayed changes in muscle-fibre organisation. Genome-wide transcriptomics revealed a general anteriorization of gene expression, and *in situ* hybridization showed an overproduction of cells expressing head-inducing factors such as *sfrp* upon *bcat-1* knockdown. Conversely, metacestode-specific genes including the tegumental factors *muc-1*, *TNFR*, and antigen B as well as the posterior marker *post2b* were significantly downregulated, consistent with the observed vesicle-formation defects. *In situ* analyses further identified anterior markers *frizzled-10*, *nou-darake*, *notum*, and *follistatin* that were overexpressed in *bcat-1*(RNAi) cultures and localized to the future anterior pole at the earliest stages of protoscolex formation. Together, these findings establish a central role for cWNT signalling in directing *Echinococcus* body-axis formation and the posteriorization events driving metacestode growth within the host, providing insight into asexual parasite proliferation mediated by this biologically unique larval stage and pointing to potential targets for chemotherapy against AE.

**Data availability statement:** All data to replicate the findings are provided in the manuscript and Supporting Information.

**Funding:** This work was funded in part by the Wellcome Trust grants 206194 to MaB and 107475/Z/15/Z to KB and MaB as well as a grants from the Wellhöfer foundation (Schwarzenbruck, Germany; grant 824000 to KB). MH and KR were supported by personal grants of the German Excellence Initiative to the Graduate School of Life Sciences (GSLS), University of Würzburg, MH_2017_013 and KR_2022_023, respectively. The funders had no role in study design, data collection and analysis, decision to publish, or preparation of the manuscript.

**Competing interests:** The authors have declared that no competing interests exist.

## Author summary

Alveolar echinococcosis (AE) is a lethal disease caused by the cancer-like growth of the metacestode larva of the tapeworm *Echinococcus multilocularis*. From a developmental perspective, the *Echinococcus* metacestode is an unusual biological structure and even atypical among tapeworms. Previous work indicated that metacestode formation involves re-patterning of the body axis, eliminating head structures and producing broadly posteriorized tissue. How this is controlled at the molecular and cellular levels, however, was unknown. In this study, we perturbed expression of the β-catenin gene (*bcat-1*), a central regulator of canonical WNT signalling, using RNA interference (RNAi). *bcat-1*(RNAi) parasite cultures failed to generate metacestode vesicles and instead showed stem-cell hyperproliferation and muscle-cell alterations. Genes required for posteriorized metacestode tissue were downregulated, whereas genes directing head formation in adult worms (*follistatin, sfrp, fz10, ndk*) were upregulated, indicating a general anteriorization of the culture system. These findings identify β-catenin and the canonical WNT pathway as crucial regulators of the posteriorization that underlies metacestode formation. Given that WNT signalling is deregulated in many human cancers and that small-molecule inhibitors are available, our results suggest new avenues for anti-AE drug development.

## Introduction

The metacestode larval stage of the fox tapeworm *Echinococcus multilocularis* causes alveolar echinococcosis (AE), a lethal zoonosis prevalent across the Northern Hemisphere [1]. In the intermediate host, parasite development is exclusively driven by a population of undifferentiated stem cells termed germinative cells (GC) [2,3]. A small number of GC enter the host as part of the oncosphere contained within infectious eggs, which are shed by the definitive host and ingested by the intermediate host [3]. After hatching in the intestine and crossing the intestinal wall, the oncosphere reaches the liver, where GC drive a metamorphosis into the metacestode: a network of fluid-filled vesicles that infiltratively grows and expands into surrounding tissue, akin to a malignant tumor [4,5]. Proliferation is exclusively GC-driven; as in other flatworm stem cell systems, these are the only mitotically active cells and give rise to all differentiated lineages (e.g., muscle, nerve, and glycogen/lipid-storing cells) [2] (S1 Fig). In natural intermediate hosts (rodents), GC in the germinal layer ultimately generate brood capsules, culminating in the formation of protoscoleces, which are juvenile, unsegmented forms that infect the definitive host upon predation [5]. In human infections, by contrast, protoscolex formation is rare [6]. Human AE is generally difficult to treat; albendazole/mebendazole chemotherapy targeting parasite β-tubulin often requires prolonged (sometimes lifelong) administration and is associated with adverse effects [7].

In typical tapeworm development, the oncosphere develops directly into scolex-bearing structures, such that a single invading oncosphere yields one juvenile

metacestode with a single scolex [5]. By contrast, the *Echinococcus* metacestode is a highly derived structure that can generate numerous protoscoleces from a single oncosphere, enabling asexual multiplication within the intermediate host [8]. As shown previously [5], metacestode formation reflects a modification of the primary anterior–posterior (A–P) axis during the oncosphere–metacestode transition. Upon entry into the intermediate host, the oncosphere appears to transiently suppress the anterior pole, marked by expression of the WNT antagonist *sfrp* (secreted frizzled-related protein), and subsequently develops into a fully posteriorized metacestode tissue broadly expressing the posterior markers *wnt1*, *wnt11a*, and *post-2* [5]. Late in infection, the anterior pole is re-established at multiple sites within the metacestode, giving rise to numerous protoscoleces that express *sfrp* anteriorly and *wnt1* posteriorly [5] (S1 Fig). This mirrors axis-formation mechanisms in related free-living planarians, where anterior and posterior poles are likewise marked by *sfrp* and *wnt1*, respectively [9]. Thus, establishment of the A–P axis is highly conserved among flatworms and, as shown in planarians [10], is under the direct control of WNT signaling, whose central component is the transcriptional regulator β-catenin [11]. Accordingly, RNAi-mediated knockdown of β-catenin in planarians yields strikingly anteriorized phenotypes: (a) during regeneration, heads form at wounds that would normally regenerate tails; and (b) during homeostatic tissue turnover, ectopic heads arise along the body axis [10,12,13]. Maintenance and regeneration of the correct body plan in planarians are therefore governed by specific WNT ligands expressed in posterior-to-anterior gradients and WNT antagonists released from the anterior [14]. Notably, in both planarians [15] and tapeworms [5], these ligands are produced by muscle cells and act as positional control genes (PCGs) that instruct stem cells to differentiate into correctly patterned tissues [14]. In tapeworms, WNT signaling also appears to contribute to A–P axis establishment during segmentation and adult development [16].

We recently characterized β-catenin orthologs in *E. multilocularis* and identified *bcat-1* as the one-to-one ortholog of the planarian β-catenin gene involved in WNT signaling [17]. Here, we test the hypothesis that *bcat-1* (EmuJ_001007700) is a central regulator of A–P axis formation in *Echinococcus* and directs development toward the metacestode. To this end, we used a primary parasite cell culture system previously established to recapitulate the oncosphere–metacestode transition *in vitro* [18,19] (S2 Fig). Within this system, we have developed functional genomic approaches to achieve targeted knockdown by RNAi [19] and, in the present work, applied them to manipulate expression of the *bcat-1* gene. We show that *bcat-1*(RNAi) cultures fail to produce mature metacestode vesicles from stem cells and exhibit a broadly anteriorized gene-expression profile. We further demonstrate that several factors essential for metacestode function are not formed by these cells, whereas anterior pole markers are significantly overexpressed. These findings indicate that, as in planarians, canonical WNT signaling is instructive for establishing the correct A–P axis in *Echinococcus* and identify β-catenin as a central regulator of metacestode formation and growth.

## Results

### *bcat-1*(RNAi) impairs metacestode development

We previously showed that the *Echinococcus* metacestode constitutes broadly posteriorized tissue, with widespread expression of flatworm posterior markers such as *wnt1* and silencing of anterior markers such as *sfrp* [5]. Because anterior–posterior (A–P) axis formation in flatworms is directed by WNT signalling [13,20], we hypothesized that this pathway acts as a master regulator of metacestode formation and asexual proliferation. The central effector of cWNT signalling is the transcriptional regulator β-catenin, and we previously identified one of the three *Echinococcus* β-catenin-encoding genes (*bcat-1*) as a likely participant in canonical WNT (cWNT) signalling in this parasite [17]. The *Echinococcus multilocularis* primary cell culture system [18] is currently the only platform that enables reliable RNAi-mediated gene knockdown and functional analyses of gene expression [19]. These cultures are highly enriched in germinative stem cells, which are the only proliferative cells and give rise to all differentiated cell types, although muscle and nerve cells are also present [2]. While mature metacestode vesicles form within 2–3 weeks, primary cell–derived stem cells also display expanded differentiation capacity and generate cell types characteristic of protoscoleces, adult worms, and oncospheres [20].

Reasoning that this system represents an ideal model to study β-catenin–dependent differentiation processes, particularly those involved in A-P axis formation, we used an established RNAi protocol for *Echinococcus* primary cell cultures [19] and targeted *bcat-1* for knockdown to assess its role in development.

As shown in Fig 1A, we achieved a stable *bcat-1* knockdown to <50% of control levels for at least 14 days in primary cell culture. In contrast to cells electroporated with control siRNAs, *bcat-1*(RNAi) cultures were markedly impaired in

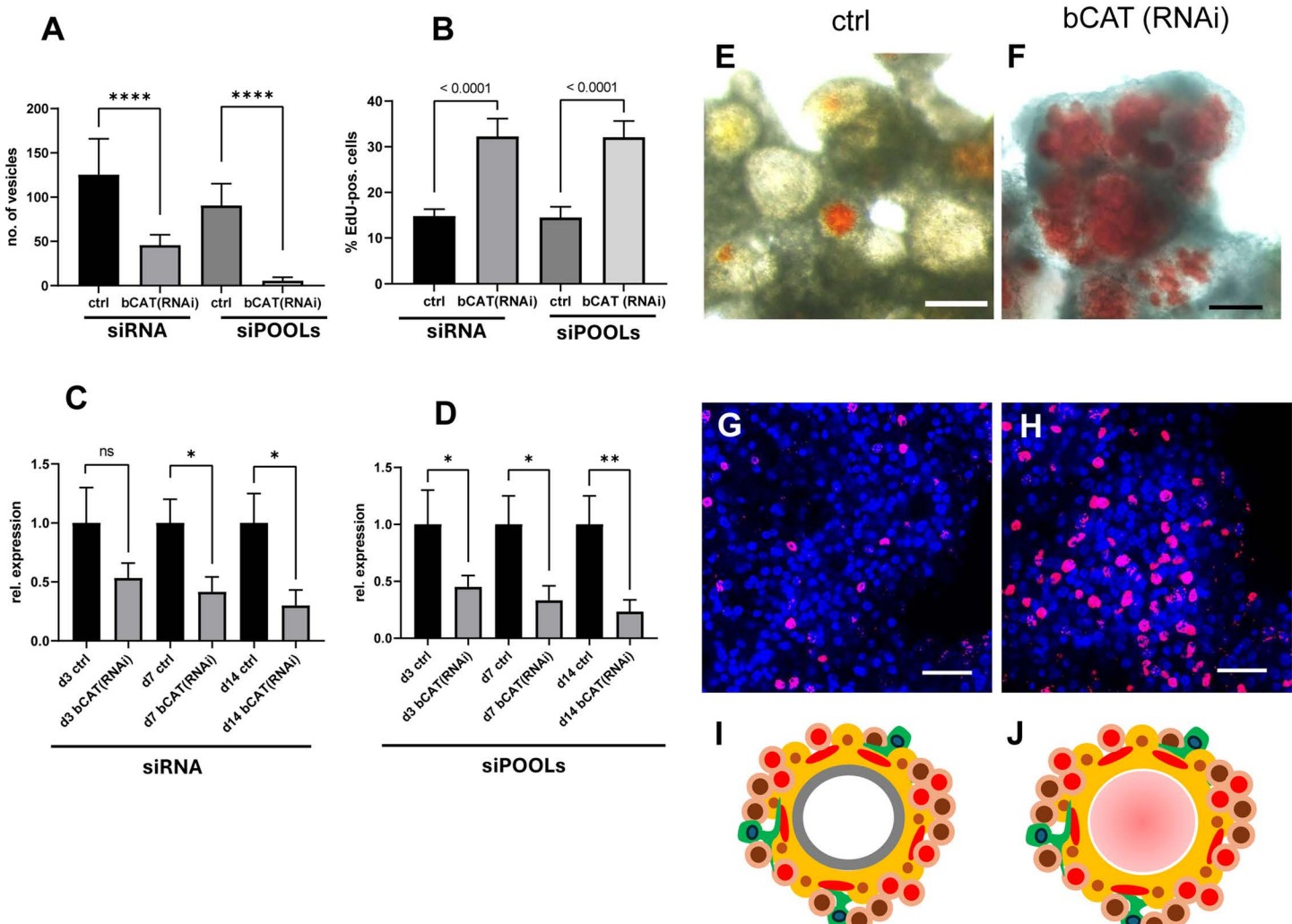

**Fig 1. *bcat-1*(RNAi) inhibits metacestode formation by *Echinococcus* cell cultures. (A)** Formation of mature metacestode vesicles by control cultures (ctrl) and *bcat-1*(RNAi) cultures (bCAT(RNAi)) after knockdown with siRNAs (siRNA) and siPOOLs (siPOOLs). Results of four biological replicates with three technical replicates each. (student's t-test, **** $p < 0.0001$). **(B)** Proportion of proliferating stem cells after 5 h EdU pulse in control (ctrl) and *bcat-1*(RNAi) cultures (siRNA and siPOOL approach as indicated) after 2 weeks of incubation. Three biological replicates with three technical replicates. (student's t-test, $p < 0.0001$). **(C) - (D)** *bcat-1* transcript levels after RNAi knockdown with siRNAs (C) and siPOOLs (D). Relative expression of *bcat-1* (grey) and control gene *elp* (black) from three biological replicates after 3 (3d), 7 (7d), and 14 (14 d) days of culture. Bars indicate standard deviation. Student's t-test, ns = not significant, * $p < 0.05$, ** $p < 0.01$. **(E)** Cell aggregates and emerging metacestode vesicles in control cultures after 2 weeks. **(F)** Cell aggregates and red dot phenotype in *bcat-1*(RNAi) cultures after 2 weeks. **(G)** Cell aggregates of control cultures (2 weeks) stained for proliferative stem cells (5h EdU pulse, red). **(H)** Cell cultures of *bcat-1*(RNAi) cultures (siRNA approach) stained for proliferative stem cells (5h EdU pulse, red). Size bar represents 100 μm in (E) and (F), 20 μm in (G) and (H). Channels in G - H were blue (DAPI, nuclei), red (EdU proliferating stem cells). **(I)** and **(J)** represent parasite primary cell cultures of control (I) and *bcat-1*(RNAi) (J) as depicted and explained in S2 Fig.

metacestode vesicle formation: after 14 days, control cultures yielded ~120 mature vesicles, whereas *bcat-1*(RNAi) cultures produced only ~40 vesicles. In controls, aggregates initially contained red-stained cavities that progressively cleared during cultivation and ultimately formed mature vesicles [18,19]. By contrast, *bcat-1*(RNAi) cultures appeared arrested at the early cavity stage, with most cavities retaining a red stain from phenol red in the medium—hereafter termed the "red-dot phenotype" (Fig 1F).

To independently validate the knockdown and minimize off-target effects, we employed the siPOOL system, which uses pooled dsRNAs spanning the entire transcript [21]. This approach again produced a stable reduction of *bcat-1* expression to <50% for ≥14 days (Fig 1D) and recapitulated the intense red-dot phenotype with developmental arrest at the cavity stage. Under these conditions, formation of mature vesicles was almost completely abolished (Fig 1A).

To investigate the cellular basis of the red-dot phenotype, *bcat-1*(RNAi) aggregates (ds RNA approach) were fixed and stained for muscle and nerve cells [22], and EdU incorporation was used to assess stem-cell proliferation. As shown in Figs 1B,1G,1H and S3, *bcat-1*(RNAi) cultures of both knockdown approaches consistently (and statistically significantly) displayed stem-cell hyperproliferation: ~32% of cells were in S-phase versus ~15% in controls. Moreover, muscle fibre arrangement in *bcat-1*(RNAi) aggregates was altered and fibres only rarely accumulated around central cavities, a pattern typically observed during vesicle formation in this system (Fig 2). Of note, muscle fibre dysregulation has also been observed after β-catenin knockdown in *Hymenolepis* diminuta [23] and *Fasciola hepatica* [24].

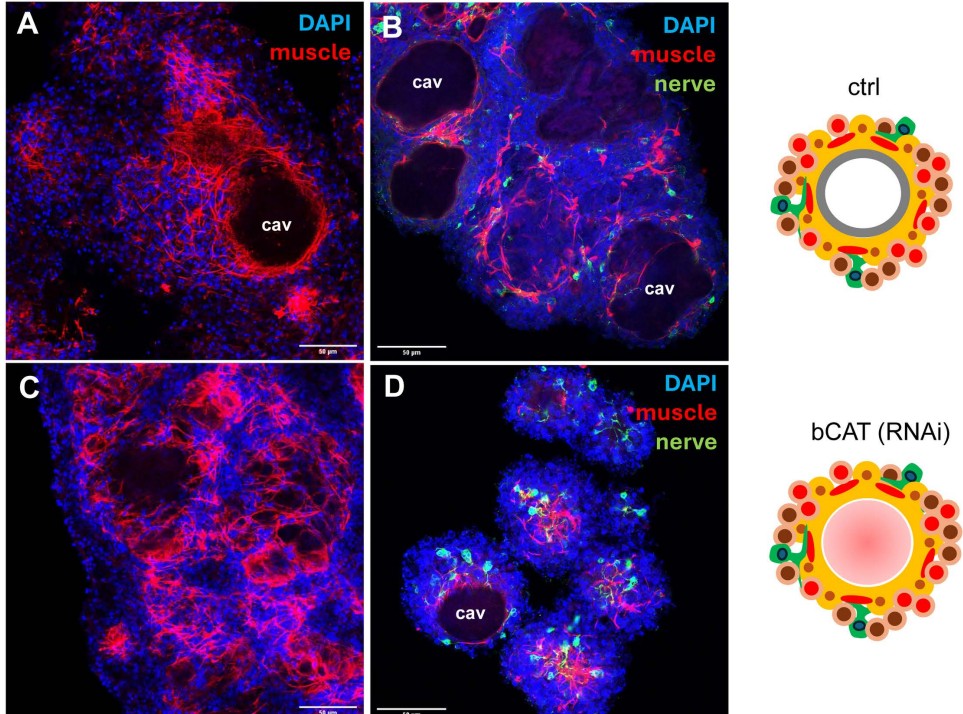

**Fig 2. *bcat-1*(RNAi) alters muscle fibre arrangement in *Echinococcus* cell cultures. (A)** Cell aggregates of 2-week-old control cultures (siRNA approach) stained for nuclei (blue) and muscle fibres (red). **(B)** Cell aggregates of control cultures (2 weeks; siRNA approach) stained for nuclei (blue), muscle fibres (red), and nerve cells (green). **(C)** Cell aggregates of 2-week-old *bcat-1*(RNAi) cultures stained for nuclei (blue) and muscle fibres (red). **(D)** Cell aggregates of *bcat-1*(RNAi) cultures (2 weeks) stained for nuclei (blue), muscle fibres (red), and nerve cells (green). Images to the right represent parasite primary cell cultures of control (ctrl) and *bcat-1*(RNAi) as detailed in S2 Fig. Scale bar represents 50 μm in (A) - (D). Channels in A - D were blue (DAPI, nuclei), red (phalloidin, muscle fibres), green (α-acTub, nerve cells). cav = central cavity.

Taken together, *bcat-1* knockdown led to a pronounced reduction in metacestode vesicle formation from *Echinococcus* primary cells, accompanied by stem-cell hyperproliferation and altered muscle-fibre organization.

### *bcat-1*(RNAi) cultures display anteriorized gene-expression patterns

To further characterize primary cell cultures after *bcat-1* knockdown, we performed genome-wide transcriptome profiling on control aggregates and *bcat-1*(RNAi) cultures (siRNA approach) after 7 days of cultivation (i.e., when the red-dot phenotype was evident). In total, 1,314 genes (12.3%) were significantly upregulated and 1,513 (14.2%) were downregulated in *bcat-1*(RNAi) cultures relative to controls (p < 0.05; S1 Table). Consistent with qRT-PCR, read counts confirmed a significant reduction of *bcat-1* expression to ~40% of control levels (S1 Table).

Given the observed stem-cell hyperproliferation in *bcat-1*(RNAi) cultures, we asked whether stem-cell-associated genes were over-represented among induced transcripts. Using a previously defined set of 1,853 stem-cell-associated genes (reduced in hydroxyurea-treated vesicles) [20], we found that 661 (35.7%) were significantly upregulated after *bcat-1*(RNAi), whereas only 139 (7.5%) were downregulated (S1 Table).

Because *bcat-1*(RNAi) was expected to affect body-axis formation, we also generated datasets for the protoscolex (PS), which exhibits an A–P axis [5] (S1 Fig), and for metacestode vesicles devoid of brood capsules, which represent posteriorized tissue without functional A-P axis [5] (S1 Fig). We defined genes significantly enriched in PS versus metacestode (MC) as PSe (PS-enriched) and those enriched in MC as MCe (MC-enriched). Because positional control genes (PCGs) in flatworms are largely associated with differentiated muscle and nerve cells [15], we focused on genes not enriched in stem cells (i.e., excluding the earlier 1,853 set). Among 651 genes upregulated after *bcat-1* knockdown, 364 were also differentially expressed between PS and MC; of these, 261 (72%) were PSe and 103 (28%) were MCe. Conversely, of 1,374 genes downregulated after *bcat-1*(RNAi), 941 were differentially expressed between PS and MC; of these, 255 (27%) were PSe and 686 (73%) were MCe. Thus, upregulated genes after *bcat-1*(RNAi) were roughly three times more likely to be PSe than MCe, whereas downregulated genes were roughly three times more likely to be MCe than PSe (Fig 2B). Correlation analysis further supported this trend, revealing a significant association between PSe status and upregulation in *bcat-1*(RNAi) cultures (Pearson's r = 0.4546, p < 0.0001; Fig 3C).

We next examined the induced gene set in more detail (Fig 3A and S1 Table). Notably, it included *sfrp* (EmuJ_000838700; 3.3-fold induced), a strict anterior-pole marker previously identified as a head-inducing factor [5], suggesting that overexpression of head-inducing programmes underlies the general anteriorization of *bcat-1*(RNAi) cultures. Among the most induced transcripts (~10-fold) was *frizzled-10* (*fz10*; EmuJ_000085700), a homolog of planarian frizzled receptors with anterior expression domains [25]. Additional WNT-pathway components upregulated after *bcat-1*(RNAi) included homologs of pangolin/TCF-LEF (EmuJ_000188500), dishevelled (*dsh*; EmuJ_000423700), and a second frizzled receptor, *fz1* (EmuJ_000682100). We also identified modulators of WNT signalling among the highly induced genes, such as *SERPINA3K* (EmuJ_000824000), which inhibits canonical WNT in *Xenopus* [26], *Glypican-1* (EmuJ_000544900), a regulator of WNT/HH/FGF/BMP pathways [27], and the homeobox factor *aristaless* (EmuJ_000574000), reported as a wound-induced negative regulator of WNT signalling [28].

In addition to posteriorly acting cWNT signalling, the planarian A–P axis is critically shaped by anterior FGF components, notably the FGF receptor-like morphogen *nou-darake* (*ndk*) and several *ndl* (nou-darake-like) factors [14]. The *E. multilocularis* genome encodes three such genes (EmuJ_000770900, EmuJ_000816800, and EmuJ_000842900; S4 Fig), and all three were 3- to 4-fold induced after *bcat-1*(RNAi) (Fig 3A and S1 Table). Likewise, both *Echinococcus* FoxD homologs (EmuJ_000620400; EmuJ_000446500), required for anterior-pole regeneration in planarians [29], were induced, as was a homolog of the morphogenic transcription factor *hunchback* (EmuJ_000448000), anteriorly expressed in *Drosophila* [30]. Altogether, the *bcat-1*(RNAi) expression profile reveals broad upregulation of WNT- and FGF-pathway genes that, in planarians and other metazoans, function as head organizers or anterior PCGs.

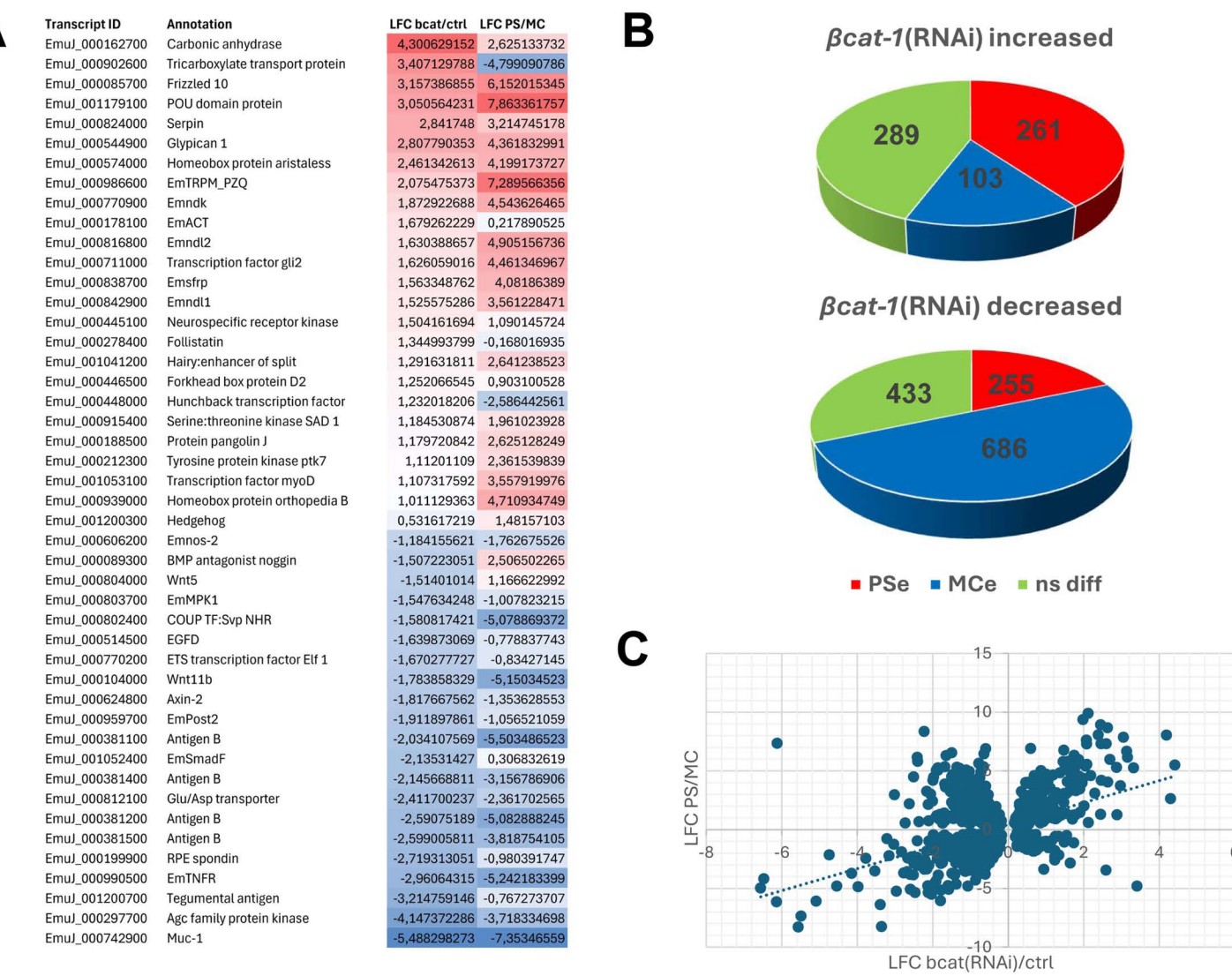

**Fig 3. *bat-1* knockdown induces anteriorization of *Echinococcus* cell cultures. (A)** Heatmap depicting transcriptional changes of selected genes upon *bcat-1*(RNAi)(si RNA approach). Indicated are the gene IDs of selected *E. multilocularis* genes alongside annotation. 'LFC bcat/ctrl' indicates average log-fold change of *bcat-1*(RNAi) cultures versus control cultures, 'LFC PS/MC' indicates log-fold changes of PSe versus MCe genes. Positive values are marked in blue, negative values in red. **(B)** Pie charts indicating relations between differential transcription profiles of genes that are up/down-regulated after *bcat-1*(RNAi)(as indicated) and PSe (red) or MCe (blue) genes. ns diff (green) indicates genes that are not significantly differentially expressed between PS and MC. **(C)** Correlation analysis plot visualizing relationships between genes that are up/down-regulated after *bcat-1*(RNAi) (x-axis) and genes enriched in PS versus MC (y-axis) (Pearson's r = 0.46, p < 0.0001).

Beyond WNT and FGF, *bcat-1*(RNAi) cultures also overexpressed components of other axis-patterning systems. These included homologs of *hedgehog* ligands (EmuJ_001200300) and a Gli-like transcription factor (EmuJ_000711000), TGFβ signalling components *emact* (EmuJ_000178100) [31] and *follistatin* (*fst*) (EmuJ_000278400; S5 Fig), and Delta/Notch components identified recently [32], such as jagged-like ligands (EmuJ_000392700; EmuJ_000851300) and a hairy/enhancer-of-split-like repressor (EmuJ_001041200). Notably, among the most strongly (>20-fold) induced transcripts were genes encoding EG95 (EmuJ_000710400; EmuJ_000368620) and gp50 antigens (EmuJ_000512300; EmuJ_000049700; EmuJ_000295100), whose cellular functions in *Echinococcus* remain unknown.

As a quality control, we validated several up- and downregulated targets by qRT-PCR in independently performed *bcat-1*(RNAi) cultures and confirmed significant up- or down-regulation in all cases (S6 Fig). Finally, *in situ* hybridization for *ndk* (EmuJ_000770900), *fz10*, *sfrp*, and *fst* revealed substantially more gene-positive cells in *bcat-1*(RNAi) cultures than in controls (S7 Fig), indicating that elevated read counts primarily reflect an increased number of expressing cells, rather than uniform per-cell upregulation.

In summary, *bcat-1* knockdown drives a pronounced shift toward protoscolex-enriched programmes and robust induction of putative anterior axis regulators and head organizers, while metacestode-enriched genes are broadly reduced. These transcriptomic changes align with the cellular phenotypes observed after *bcat-1*(RNAi).

### *bcat-1*(RNAi) primary cells fail to express key factors required for metacestode development

We next examined genes that were significantly repressed in primary cells following *bcat-1*(RNAi). As shown in S1 Table, one of the most strongly reduced transcripts was *muc-1* (EmuJ_000742900; >40-fold reduced), which encodes a member of an *Echinococcus*-specific apomucin family and a major component of the laminated layer [33]. In our primary cell culture system, we recently demonstrated that metacestode vesicles initially arise as inside-out vesicles (central cavities encircled by *muc-1*–positive tegumental structures) from which mature, outside-in vesicles later develop [20] (S2 Fig). By *in situ* hybridization, we confirmed this here for RNAi control cultures (Fig 4A), which showed intense *muc-1* staining around central cavities. In *bcat-1*(RNAi) cultures, by contrast, *muc-1* expression around cavities was rarely detected and was in most aggregates entirely absent, despite comparable aggregate size (Fig 4B).

A second, as yet uncharacterized mucin gene, *muc-2* (EmuJ_000938200; S8 Fig), was likewise >13-fold repressed after *bcat-1*(RNAi). In controls, *muc-2* strongly labelled a cellular layer adjacent to central cavities, whereas no staining was observed in *bcat-1* knockdown aggregates (Fig 4B and 4D). Beyond mucins of the laminated layer, another hallmark of *Echinococcus* metacestode vesicles is robust expression of antigen B lipoproteins, major constituents of hydatid fluid [34]. These isoforms are encoded by a cluster of five genes in the *Echinococcus* genome [35], and all five were strongly suppressed after *bcat-1*(RNAi) (S1 Table).

Additional metacestode-specific factors with significantly reduced expression included a tumour necrosis factor receptor family member (*TNFR*; EmuJ_000990500) recently described by us [20]; the WNT morphogen *wnt11b* (EmuJ_000010400); the posterior marker *post2b* (EmuJ_000959700) [5]; and an axin ortholog (*axin-1*; EmuJ_000624800), previously shown to be expressed in the metacestode and in posterior regions of the developing protoscolex [17]. Notably, among the most strongly reduced transcripts we also identified an ortholog of ribosomal protein S6 kinase 1 (*S6K1*; EmuJ_000297700), which has been reported to enhance oncogenic WNT signalling by promoting formation of the β-catenin transcriptional complex in HEK293T cells [36], as well as an uncharacterized inhibitory Smad (iSmad) family member (*smadF*; EmuJ_001052400; S9 Fig).

As with *bcat-1*(RNAi)-induced genes, we validated repression by *in situ* hybridization: *muc-1*, *muc-2*, *post2b*, *wnt11b*, *TNFR*, and *smadF* were readily detected in control cultures but were strongly reduced or absent in *bcat-1*(RNAi) aggregates (Figs 4 and 5).

Taken together, these analyses show that multiple hallmark components crucial for metacestode vesicle development and physiology are profoundly suppressed in primary cell cultures upon *bcat-1* knockdown.

### FGF- and WNT-signalling head organizers are induced during protoscolex development

Given (i) our previous observation of clear homologies in gene-expression patterns between cestode larval metamorphosis and late planarian development [5], and (ii) the fact that many *bcat-1*(RNAi)-induced genes belong to the planarian positional-control toolkit that specifies body axes [37], we hypothesized that these genes are activated at the onset of brood-capsule formation and during protoscolex development. We further posited that their expression domains within the protoscolex would mirror those of their planarian homologs. To test this, we performed WISH on metacestode vesicles across defined stages of protoscolex production (see S1 Fig for comparison).

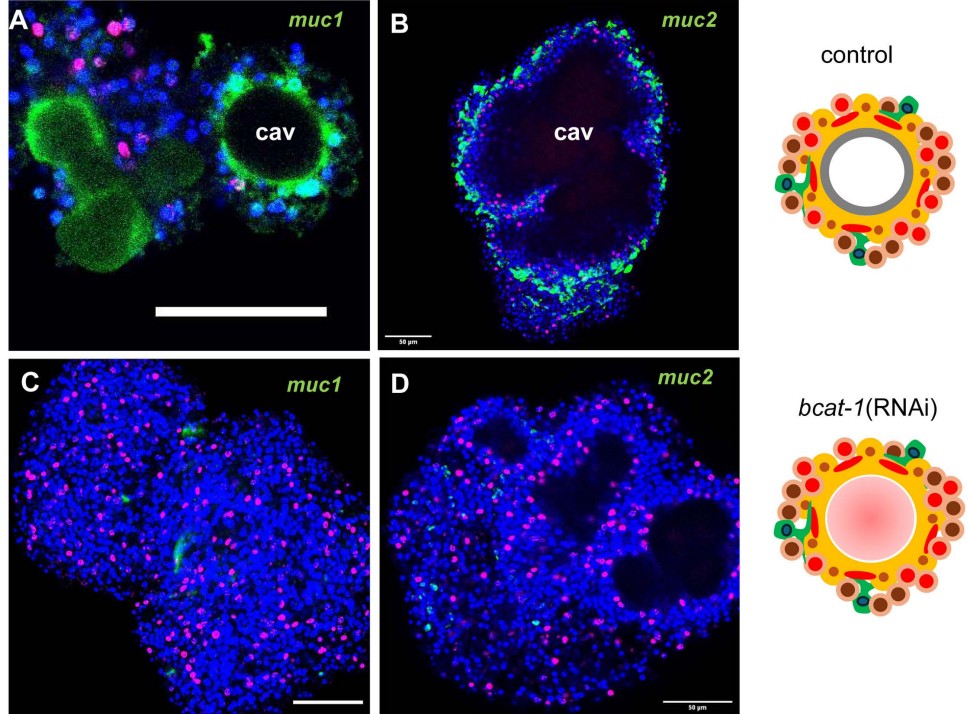

**Fig 4. *bcat-1*(RNAi) cultures fail to express metacestode-specific mucin genes.** Shown are representative WISH analyses for selected *E. multilocularis* genes (indicated above to the right) in cell aggregates of control cultures (control; A,B) and *bcat-1*(RNAi) cultures (bcat-1(RNAi); C,D). Shown are merge pictures (single confocal slices) of all three channels: blue (nuclei, DAPI), red (proliferative cells, EdU+), green (gene-specific probe). Scale bar represents 50 μm in all panels. cav, central cavities. Images to the right represent parasite primary cell cultures of control (ctrl) and *bcat-1*(RNAi) as detailed in S2 Fig.

For *fz10*, no expression was detected in metacestode vesicles lacking brood capsules (Fig 6A), but clear signals appeared at the onset of brood-capsule formation, centred on the cell accumulations that mark early capsule development (Fig 6B). Throughout protoscolex development, *fz10* was expressed in two domains immediately lateral to the anterior pole (Fig 6C-6E), whereas expression ceased in fully developed protoscoleces (Fig 6F). Based on this pattern and on the strong repression of *fz10* under functional WNT signalling we conclude that *fz10* is likely among the earliest *Echinococcus* genes acting as an anterior head inducer to promote protoscolex formation; its expression appears dispensable once the larva is fully formed.

Similarly, *ndk* was undetectable in vesicles without brood capsules (Fig 7I), then became induced in early brood capsules and remained close to the anterior pole until protoscolex development was complete (Fig 7A-7D). It should be noted that prior to the formation of the rostellum (hooked structure at the anterior end), *ndk* was still expressed at the anterior pole, whereas in fully mature protoscoleces we observed a collar-like expression pattern surrounding the rostellum (Fig 7A-7D). Unlike *fz10*, *ndk* expression persisted after protoscolex completion. *ndl1* and *ndl2* were likewise not expressed in metacestode vesicles (Fig 7J and 7K) but induced early near the anterior pole at brood capsules (Fig 7E). In mature protoscoleces, *ndl2* formed a collar-like domain adjacent to *ndk* but slightly more posterior (Fig 7G and 7H), and *ndl1* was expressed in a domain around the suckers, immediately posterior to *ndl2* (Fig 7E and 7F).

Because orthologs of *fz10*, *ndk*, *ndl1*, and *ndl2* pattern anterior domains in planarians [38], we infer that these genes are strongly repressed in metacestode vesicles during infiltrative growth but become induced at the onset of brood-capsule formation to establish a head-organizing centre maintained throughout protoscolex development. Whereas *fz10*

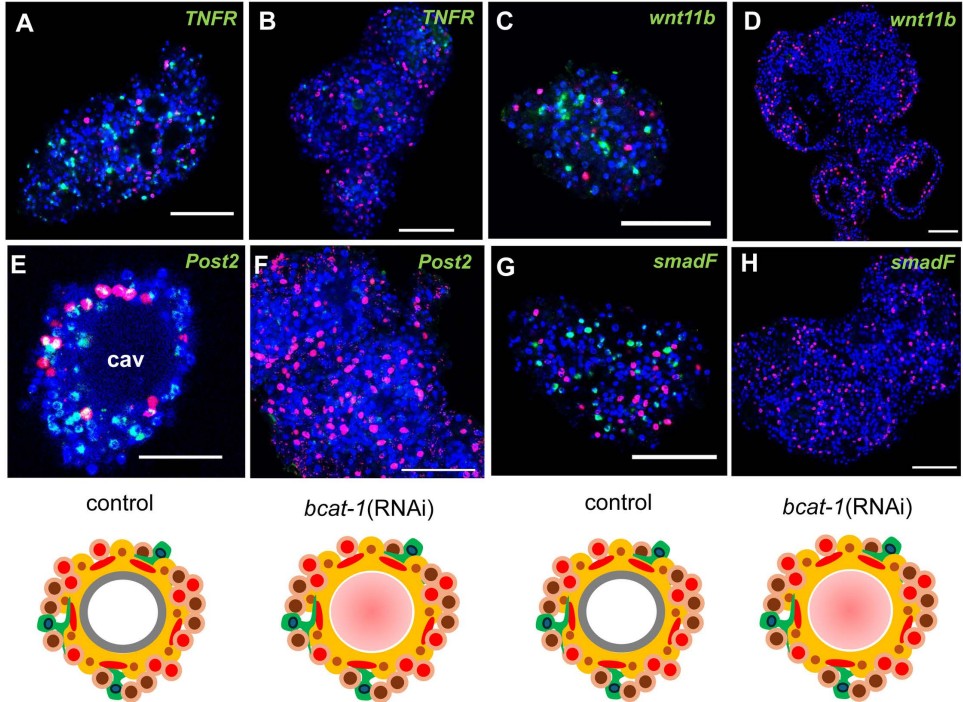

**Fig 5. *bcat-1*(RNAi) cultures fail to express metacestode markers.** Shown are representative WISH analyses for selected *E. multilocularis* genes (indicated above to the right) in cell aggregates of control cultures (control; A,C,E,G) and *bcat-1*(RNAi) cultures (bcat-1(RNAi), B,D,F,H). Shown are merge pictures (single confocal slices) of all three channels: blue (nuclei, DAPI), red (proliferative cells, EdU+), green (gene-specific probe). Scale bar represents 50 μm in all panels except E where it is 20 μm. Images below represent parasite primary cell cultures of control (control) and *bcat-1*(RNAi) as detailed in S2 Fig.

is no longer expressed once the protoscolex is complete, the FGF-signalling components *ndk*, *ndl1*, and *ndl2* remain expressed in the anterior region.

### *notum* and TGF-β signalling components are expressed in the metacestode and anteriorly during protoscolex formation

Previous work has shown that re-establishment of the anterior pole during planarian regeneration requires the WNT antagonist *notum* and the TGF-β signalling components *follistatin* (*fst*) and *activin* [39,40]. We previously found that an *Echinococcus* activin ortholog (*emact*; EmuJ_000178100) is expressed in the metacestode during infiltrative growth and is strongly upregulated near the anterior pole during brood-capsule and protoscolex formation [41]. Consistent with this, *emact* was induced more than three-fold in *bcat-1*(RNAi) cultures relative to controls (Fig 3A and S1 Table). We also identified a single *fst* ortholog in the *E. multilocularis* genome (EmuJ_000278400; S5 Fig), which, like *emact*, was induced about three-fold in *bcat-1*(RNAi) cultures (Fig 3A and S1 Table). By contrast, the only *notum* ortholog in the genome (EmuJ_000652800; S10 Fig) was not induced after *bcat-1*(RNAi) (S1 Table). To assess the involvement of these factors in anterior-pole re-establishment in *Echinococcus*, we therefore performed additional in situ hybridizations for *notum* and *fst*.

As shown in Fig 8, both *notum* and *fst* were expressed in post-mitotic cells throughout the germinal layer of vesicles lacking brood capsules. *notum* transcripts were detected in 4.4±0.5% of all germinal-layer cells, whereas 9.9±1.0% stained positive for *fst*. Double-positive cells for the gene-specific probe and EdU were rare (< 0.1% for both genes),

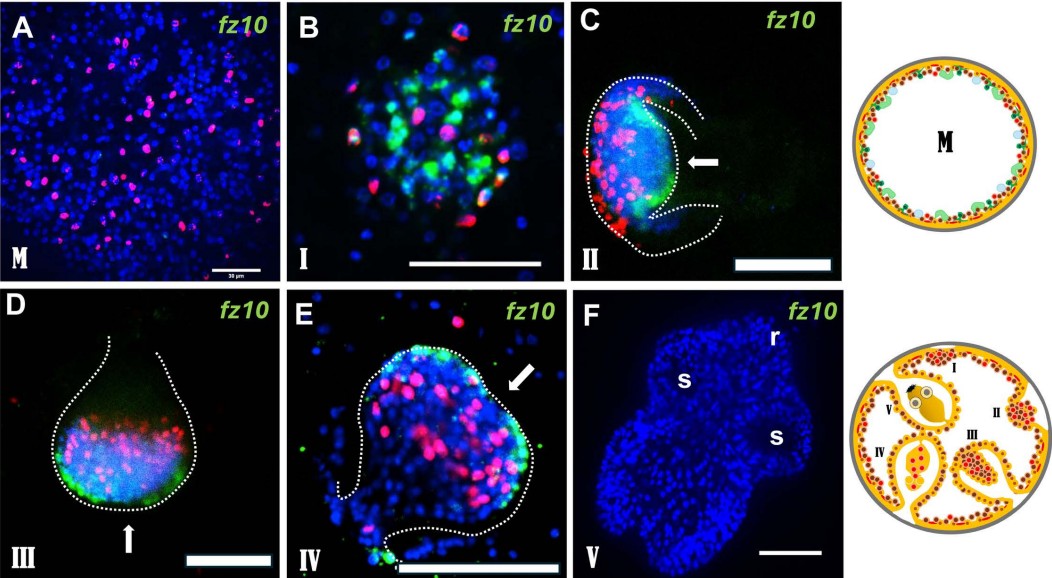

**Fig 6. _fz10_ is among the earliest genes that are expressed in brood capsules and anteriorly during PS development. (A)** Lack of _fz10_ expression in metacestode vesicles without brood capsules. **(B) – (E)** Expression of _fz10_ in early brood capsule (B; stage I) and during different phases of PS development (C – E; phases are indicated to the lower left of the images). **(F)** Lack of _fz10_ expression in fully developed protoscolex. Dashed lines indicate brood capsule structure. Note the expression domains lateral to the anterior pole (arrow). r indicates rostellum, s indicates suckers. All images are merge pictures (single confocal slices) of three channels: blue (nuclei, DAPI), red (proliferating stem cells, EdU+), green (gene specific probe). Scale bars are 30 µm in A,B,F and 50 µm in C,D,E. Images to the right represent metacestode without brood capsule (above; M) as well as broodcapsule/protoscolex development in metacestode vesicles at different stages (below; I – V) parasite primary cell cultures as detailed in S1 Fig.

indicating that expression is confined to post-mitotic cells. For _fst_ we observed expression in cells immediately beneath the tegument (Fig 5G) at a location where muscle cells are found [22].

At the onset of brood-capsule formation, _notum_ expression appeared in single cells at the anterior pole and remained detectable there throughout protoscolex development (Fig 8B-8D). In almost fully developed protoscoleces, a faint _notum_ signal persisted near the rostellum (Fig 8E). _fst_ likewise showed strong induction at the start of brood-capsule formation (Fig 8H). During protoscolex development, _fst_ was expressed in three domains: two lateral to the anterior pole and one extending centrally from the anterior to the posterior pole (Fig 8I), closely resembling the pattern of the FGF-receptor gene _emfr2_ [42]. In mature (activated) protoscoleces, _fst_, like _emfr2_, remained expressed within and around the sucker region (Fig 8J).

Thus, in contrast to _fz10_, _ndk_, _ndl1_, and _ndl2_, which are absent from metacestode tissue and become strongly induced only upon brood-capsule formation, _emact_ [41], _notum_, and _fst_ display expression profiles consistent with roles in anterior-pole formation while also being broadly expressed in post-mitotic cells of the germinal layer in vesicles that have not yet initiated brood-capsule development.

## Discussion

A defining feature of tapeworms of the genus _Echinococcus_ is the formation of large metacestodes arising from extensive growth of the germinal layer and enabling prolific asexual multiplication in the intermediate host. Our previous work suggested that metacestode development reflects extensive modification of larval body axes during the oncosphere-to-metacestode transition [5]. We showed that protoscoleces display opposing poles of cells expressing posterior WNT ligands (e.g., _wnt1_) and anterior WNT antagonists (e.g., _sfrp_), indicating that fundamental mechanisms of anteroposterior axis formation known

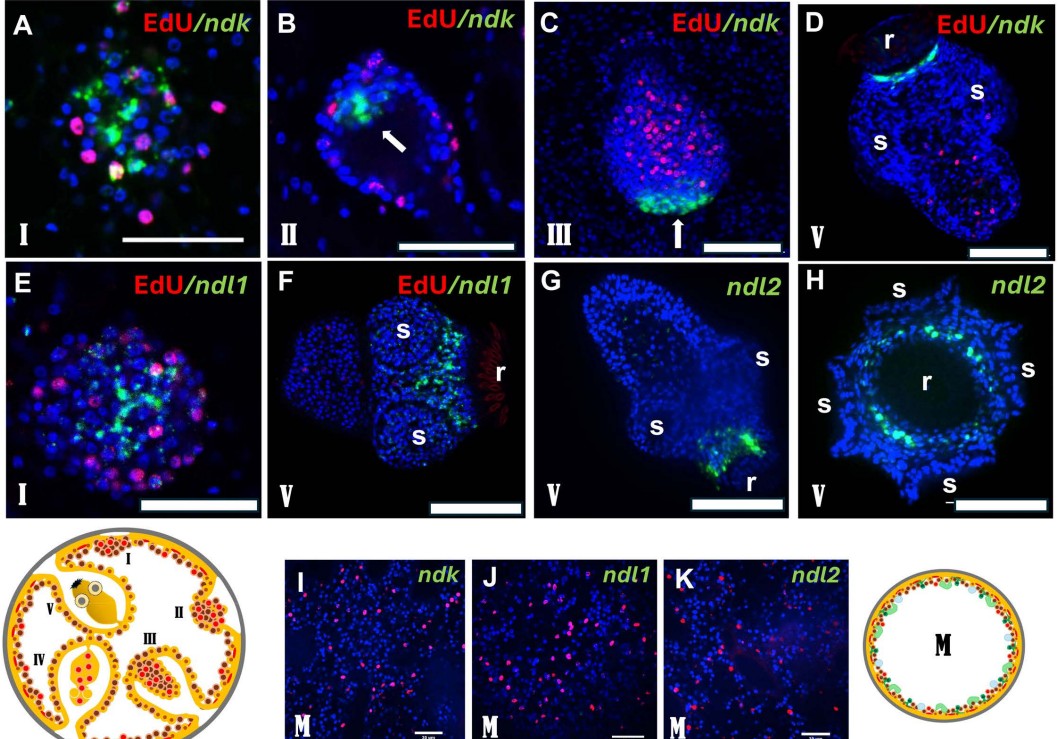

**Fig 7. FGF-signalling components are expressed in brood capsules and anteriorly during PS development. (A) – (D)** Expression of *ndk* in early brood capsules **(A)**, the developing PS **(B,C)**, and mature PS **(D)**. Arrow indicates anterior pole, r indicates rostellum, s indicates suckers. **(E) – (F)** Expression of *ndl1* in early brood capsule (E) and mature PS **(F)**. r indicates rostellum, s suckers. **(G) – (H)** Expression of *ndl2* in mature PS. (G) side view, (H) view from anterior pole. r indicates rostellum, s suckers. **(I,J,K)** Lack of expression of *ndk*, *ndl1*, and *ndl2* (as indicated) in metacestode vesicles without brood capsules. All images (except G,H) are merge pictures (single confocal slices) of three channels: blue (nuclei, DAPI), red (proliferating stem cells, EdU), green (gene specific probe). **(G) – (H)** are merge pictures of DAPI and gene specific probe only. Scale bars are 30 μm in A,B,E,I,J,K and 50 μm in C,D,F,G. Image below left represents broodcapsule/protoscolex development in metacestode vesicles at different stages (I – V) as detailed in S1 Fig. Image below right indicates metacestode vesicle without brood capsule as described in S1 Fig. Respective stages (M, I – V) are indicated in individual confocal slices.

from free-living flatworms [9] also apply to parasitic species. In sharp contrast, the *Echinococcus* metacestode constitutes fully posteriorized tissue, with *wnt1*-expressing muscle cells dispersed across the germinal layer and complete suppression of the WNT antagonist *sfrp* [5]. Expression of *sfrp* resumes later in infection at sites where brood capsules and protoscoleces form. Based on these findings, we proposed that cWNT signalling contributes to the regulation of metacestode development [5,43].

Here, we functionally interrogated cWNT signalling during developmental transitions of *E. multilocularis*. Using a primary culture system for *Echinococcus* stem cells [18,19] (S2 Fig), which recapitulate the oncosphere–metacestode transition *in vitro*, and RNA interference, we employed both short interfering RNAs and siPOOLs spanning the entire transcript. Both approaches yielded a robust and reproducible developmental phenotype (the "red-dot phenotype") clearly implicating the targeted gene, *bcat-1*, in metacestode formation. This contrasts with earlier RNAi efforts against cytoskeletal or glycolytic genes in the same system, which did not elicit detectable phenotypes [19]. We attribute the difference to the inherent instability of β-catenin: in the cytoplasm, it is rapidly degraded by the β-catenin destruction complex [13], making it highly sensitive to even modest reductions in transcript abundance. Moreover, *bcat-1* encodes the sole β-catenin of the canonical pathway in *Echinococcus* [17], precluding compensation by alternative isoforms. To our knowledge, and aside from attempts targeting protoscolex genes [44,45] or an *Echinococcus* microRNA [46], this is the first report of a robust

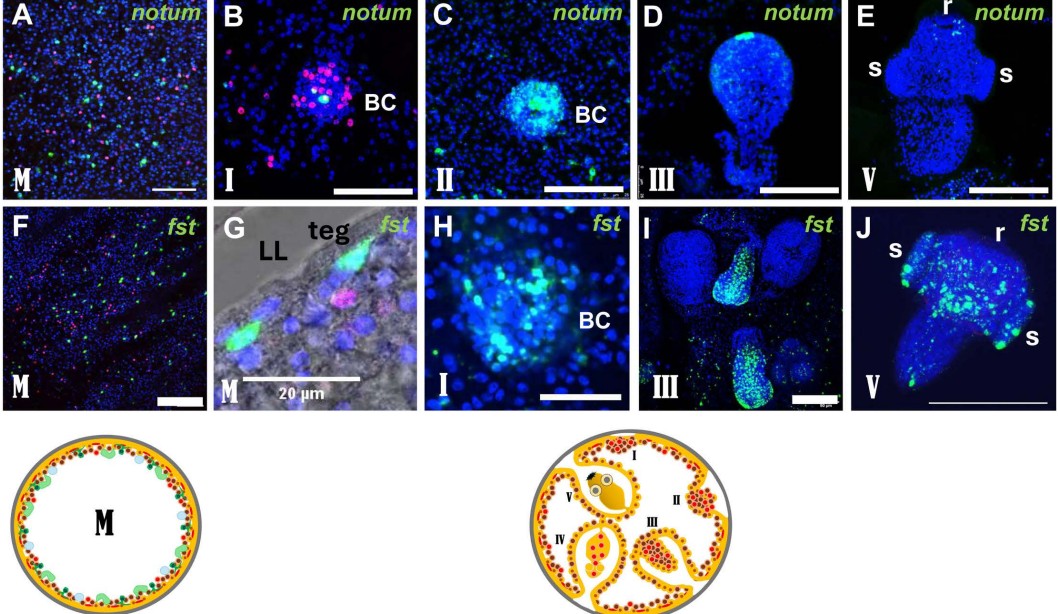

**Fig 8. Expression of *notum* and *fst* in metacestode, brood capsules, and developing PS. (A) – (E) Expression of *notum* in metacestode tissue (A)**, early brood capsule **(B,C)**, developing PS **(D)**, and nearly complete PS **(E)**.(A,B) show merge pictures of single confocal slices of all three channels (blue, DAPI, nuclei; red, EdU +, proliferating stem cells; green, gene specific probe) after 5 h EdU pulse. (C-E) show merge pictures of single confocal slices of two channels (blue, green, as in A/B). **(F – J)** Expression of *fst* in metacestode tissue **(F,G)**, early brood capsule **(H)**, developing protoscolex **(I)**, and mature protoscolex **(J)**. (F,G) show merge picture of single confocal slices of all three channels (as in A,B) after 5 h EdU pulse. (H,I) show merge pictures of single confocal slices of two channels (as in C-E). (J) shows z projection of several confocal slices of single (activated) protoscolex with two channels (blue, nuclei, DAPI; green, gene specific probe). BC, brood capsule; LL, laminated layer; teg, tegument, r, rostellum, s, sucker. Size bars represent 20 μm in G, 200 μm in J, and 50 μm in all others. Images below represent metacestode vesicle devoid of brood capsule (M) and broodcapsule/ protoscolex development in metacestode vesicles at different stages (I – V) as detailed in S1 Fig. Respective stages are indicated in confocal slices to the lower left.

phenotype following RNAi-mediated knockdown of a protein-coding *E. multilocularis* gene essential for the metacestode. Notably, we previously observed a similar red-dot phenotype after morpholino-based knockdown of microRNA *mir-71*, one cellular target of which is *fz10* [46]. The phenotypic similarity between *mir-71* and β-catenin knockdowns may thus, at least in part, reflect upregulation of a single gene, *fz10*, which we identify here as strongly induced upon WNT inhibition.

Our results reinforce the view that larval metamorphosis in tapeworms and embryonic development in planarians represent phylotypic stages within their respective flatworm clades, with cWNT signalling crucial for establishing the anteroposterior axis in tapeworms. Building on conserved expression of *wnt1* and *sfrp* at posterior and anterior poles [5], we now show that the *nou-darake*-like FGF signalling family (orthologs of mammalian FGFRL) also contributes instructively to head patterning in *Echinococcus*. Our study adds *ndk*, *ndl1*, *ndl2*, and *fz10* to the catalogue of *Echinococcus* patterning factors strongly upregulated during brood-capsule formation, alongside *sfrp* and *sfl* previously characterized [5]. Because these factors are absent from metacestode tissue, we propose that they do not trigger brood-capsule initiation; rather, they act in head patterning once the decision for brood-capsule formation has already been made. In contrast, we find three genes (*notum*, *fst*, and *activin*) to be broadly expressed in metacestode tissue before brood-capsule formation and subsequently at or near the anterior pole during protoscolex development. Importantly, these same factors act upstream of *sfrp* during re-establishment of the anterior pole in planarian regeneration. *notum* and *fst* are co-expressed in precursors of *sfrp*-positive anterior-pole cells [29,39,40], and *activin* collaborates with *notum* to set regeneration polarity [47]. We therefore propose that the interplay among *notum*, *fst*, and *activin* within metacestode tissue during infiltrative growth

determines the onset of brood-capsule formation, which is then followed by expression of *sfrp*, *fz10*, and *ndk* to drive head patterning. The molecular and cellular events by which *notum*, *fst*, and *activin* induce brood capsules remain unknown and, as in planarians, will require extensive future experimentation. Of note, we recently showed that mammalian TGF-β induces brood-capsule formation in vitro [41], suggesting that host-derived factors must also be considered. In addition to *emact* upregulation after *bcat-1*(RNAi), we observed strong downregulation of a previously uncharacterized inhibitory Smad (*smadF*). iSmads are TGF-β/BMP signalling factors that form complexes with regulatory Smads and receptor intracellular domains and, at least in mammals, can also interact with β-catenin [48,49]. Decisions governing brood-capsule induction in *Echinococcus* may therefore involve not only extracellular interactions between *fst*, *emact*, and host TGF-β but also substantial intracellular WNT/TGF-β crosstalk.A key finding of our study is that *bcat-1*(RNAi) primary cultures are profoundly compromised in forming mature metacestode vesicles. This defect coincides with significant downregulation of posterior markers (e.g., *wnt11b*, *post2*, *axin1*) and of factors essential for metacestode structure and function. Chief among these is *muc-1*, encoding the protein component of a major glycoprotein of the acellular laminated layer, which surrounds the tegument and protects the parasite from host immune cells [6,50]. We likewise observed reduced expression of a tegument-specific alkaline phosphatase isoform [2] and of all genes encoding Antigen B isoforms, which are critical for lipid uptake from the intermediate host [51]. Together, these data indicate a diminished capacity of *bcat-1*(RNAi) primary cultures to generate hydatid-fluid components and structures of the metacestode tegument, supporting the hypothesis that proper metacestode development requires functional cWNT signalling. Previously, we characterized metacestode-specific WNT ligands *wnt11b*, *wnt11a*, and *wnt1* as prominently expressed across the germinal layer of mature vesicles and at the posterior pole of protoscoleces [5]. As in planarians, these ligands likely help maintain the posteriorized identity of the metacestode [13]. Because *wnt11b* and *wnt1* are expressed by numerous cells in developing primary cultures, they likely drive stem-cell differentiation toward posterior fates necessary to establish vesicles [41]. Their marked downregulation in *bcat-1*(RNAi) cultures can thus be considered a principal cause of the developmental arrest.

In addition to impaired differentiation, *bcat-1(RNAi)* cultures exhibited increased proliferation of germinative cells. One possible explanation for this observation is that inhibition of β-catenin interferes with differentiation toward posterior cell fates, resulting in continued proliferation of stem cells that fail to undergo lineage commitment. Such a differentiation defect would be expected to result in accumulation of proliferating germinative cells and impaired tissue formation.

However, previous work suggests that increased germinative cell proliferation is also a physiological feature of anterior developmental processes in *Echinococcus* [2,5]. During brood capsule and protoscolex formation, germinative cells accumulate and undergo extensive proliferation in regions characterized by expression of anterior patterning factors and reduced WNT signalling activity [5]. These findings indicate that activation of anterior developmental programs is associated with increased stem-cell proliferation. In this context, the hyperproliferation observed following *bcat-1* knockdown may reflect not only impaired differentiation but also a shift toward an anterior developmental state. This interpretation is supported by the transcriptomic anteriorization observed in *bcat-1*(RNAi) cultures, including increased expression of anterior patterning genes such as *sfrp*, *fz10*, and related factors

Recent transcriptomic analyses of primary cell cultures further support this interpretation. These studies demonstrated that primary cell aggregates are highly enriched in germinative stem cells but retain extensive differentiation potential and express genes associated with multiple developmental stages, including metacestode, protoscolex, and adult worm stages [5,20]. Moreover, germinative cells in primary culture give rise to differentiated cell types such as muscle and metacestode tegumental cells and express key markers of metacestode development, including *muc-1*, indicating that these cultures recapitulate intrinsic parasite developmental programs [20]. These observations suggest that the primary cell system reflects physiologically relevant developmental processes and that inhibition of β-catenin may alter the balance between alternative developmental programs rather than simply blocking differentiation.

The observed anteriorization of gene expression in *bcat-1(RNAi)* cultures provides a mechanistic explanation for the impaired vesicle formation. In planarians and other flatworms, cWNT signalling promotes posterior identity and

suppresses anterior developmental programs [12,13]. Similarly, in *Echinococcus*, posterior WNT ligands are expressed throughout metacestode tissue and likely contribute to maintaining its posteriorized identity [5]. Inhibition of β-catenin signalling leads to upregulation of anterior markers and suppression of posterior-associated genes, indicating that canonical WNT signalling contributes to maintaining posterior identity during metacestode development.

An important consideration is the extent to which findings obtained in primary cell cultures reflect metacestode development *in vivo*. Primary cell cultures reproduce key aspects of metacestode formation, including vesicle morphogenesis, differentiation of germinal-layer cell types, and expression of metacestode-associated genes [2,5,18,20]. Germinative cells *in vitro* retain the capacity to proliferate and differentiate into multiple cell types present in metacestodes, including tegumental and muscle cells [5,20]. Transcriptomic analyses have shown that germinative cells express genes associated with stem-cell regulation, differentiation, and developmental patterning, supporting the relevance of this system for studying intrinsic developmental mechanisms [5,20]. At the same time, the *in vitro* system does not fully reproduce host-derived environmental signals, immune interactions, or tissue-level constraints present during infection. Therefore, while the primary cell system provides a useful model for investigating developmental processes, differences in quantitative aspects of proliferation and differentiation between *in vitro* and *in vivo* conditions cannot be excluded.

Taken together, our findings indicate that canonical WNT signalling contributes to regulating axial identity, stem-cell proliferation, and differentiation during metacestode development. Inhibition of β-catenin leads to impaired differentiation, altered axial patterning, and increased stem-cell proliferation, likely reflecting disruption of the balance between anterior and posterior developmental programs.

These findings may also have implications for therapeutic approaches. Since parasite growth is driven by germinative stem cells, signalling pathways regulating stem-cell behaviour represent potential targets for intervention. Pharmacological modulation of WNT signalling could therefore affect parasite development and viability. In this context, the clinically approved anthelmintic drug pyrvinium pamoate, which interferes with canonical WNT signalling, may represent a candidate for further investigation [24,52]. Future studies will be required to determine whether targeting this pathway can provide new therapeutic strategies for the treatment of echinococcosis.

## Materials and methods

### Parasite material and *in vitro* cultivation

Experiments were carried out using parasite isolates Ingrid, GH09, MS1010, GT10, J2012, DDD16, MB17, and TM19 that all derive from Old World Monkey species naturally infected in a breeding enclosure [53]. Isolate H95 derived from a naturally infected fox of the Swabian mountains, Germany [18]. Metacestode tissue was continuously propagated by intraperitoneal passage in Mongolian jirds (*Meriones unguiculatus*) as previously described [54]. At the time point of these experiments, all isolates except H95 were capable of brood capsule and protoscolex production both *in vitro* and *in vivo*. Isolation and cultivation of parasite primary cell cultures was performed as previously described [18,19,54]. Axenic cultivation was performed using conditioned medium from rat Reuber cells, nitrogen gas phase, and reducing conditions as previously described [60] with medium changes every 3 – 4 days. Protoscoleces were isolated from *in vivo* cultivated parasite material as previously described [22] and activated by low pH/pepsin/taurocholate treatment as described by Ritler et al. [55].

### Nucleic acid isolation, cloning, and sequencing

Total RNA was isolated from specimen using the RNEasy kit (Qiagen, Hilden, Germany) according to the manufacturer's instructions and cDNA was generated using oligonucleotide CD3-RT as previously described [20]. PCR products were cloned employing the the TOPO TA cloning kit (Thermo Fisher Scientific) and sequenced by the Sanger method using primers binding to vector sequences adjacent to the cloning sites. In cases of newly described genes, cDNA regions

spanning the entire coding region plus 5' and 3' non-translated regions, as predicted according to transcriptome data, were PCR amplified using gene specific primers (listed in S2 Table). The corrected full-length sequences of all genes newly characterized in this study were submitted to the GenBank database and are available under the accession numbers listed in S2 Table.

### *In situ* hybridization, immunohistochemistry and EdU labelling

Whole-mount *in situ* hybridization (WISH) was performed on *in vitro* cultivated metacestode vesicles or protoscoleces according to previously established protocols [2,5]. Digoxygenin (DIG)-labeled probes were synthesized by *in vitro* transcription with T7 and SP6 polymerase (New England Biolabs) using the DIG RNA labelling kit (Roche) from cDNA fragment cloned into vector pJET1.2 (Thermo Fisher Scientific). Amplification primers for all probes used in this study are listed in S2 Table. After hybridization, all fluorescent specimens were processed and analysed as described [41,56]. Control experiments using labeled sense probes were always negative. *In vitro* staining of S-phase stem cells was carried out as previously described [2] using 50 µM 5-ethynyl-2'-deoxyuridine (EdU) for a pulse of 5 h after vesicle isolation, followed by fluorescent detection with Alexa Fluor 555 azide as described previously [41,56]. For *in situ* hybridization on primary cell preparations, the established protocol for metacestode vesicles [5] was slightly modified by including additional sedimentation steps during washing to avoid material loss.

### RT-qPCR

Total RNA was isolated from primary cell aggregates as described above and cDNA was synthesized as described previously [2]. qPCR was then performed according to a previously established protocol [46,57] on a StepOne Plus Real-time PCR cycler (Applied Biosystems). Primer sequences for all genes analyzed in this study are listed in S2 Table. The constitutively expressed gene *elp* (EmuJ_000485800) was used as a control and tested for constitutive expression on control and *bcat-1*(RNAi) aggregates against the genes EmuJ_000375800 and EmuJ_000925000 (S11 Fig). Cycling conditions were 15 min at 95°C, followed by 40 cycles of 15 sec at 95°C, 20 sec of 58°C, and 20 sec of 72°C. PCR efficiencies were calculated using LinRegPCR [58], amplification product specificity was assessed by melting curve analysis and gel electrophoresis. Expression levels were calculated by the efficiency correction method using cycle threshold (Ct) values according to [57].

### RNA interference and RNA-Seq sample preparation

*bcat1*-specific RNAi was carried out on *E. multilocularis* (isolate: Ingrid) primary cells using a previously established protocol [19] and siRNAs as listed in S2 Table. siRNAs directed against green fluorescent protein were used as a control. After 7 days of incubation, when the 'red-dot-phenotype' was apparent in *bcat-1*(RNAi) cultures, samples were collected and further processed for RNA isolation. In a complimentary approach to *bcat-1* siRNAs, transient knockdown of *bcat-1* gene expression was conducted by electroporation of primary cells using siRNA pools (siTOOLs Biotech GmbH, Martinsried, Germany) which contain up to 30 siRNAs to achieve maximal transcript coverage and minimize off target effects [21], employing random siRNAs (siTOOLs) as control. In the case of protoscolex preparations, *in vivo* cultivated parasite material (isolates: MS1010, Ingrid, GH09) was isolated from Mongolian jirds [54] and protoscoleces were isolated as previously described [54]. All samples were washed three times with ice-cold PBS (phosphate buffered saline). Samples were then transferred to TRIZOL-Reagent (Invitrogen, Karlsruhe, Germany) and stored at -80°C. Tissue samples were washed with ice-cold PBS before being mechanically homogenized in TRIZOL for 10 seconds. 200 µl of chloroform:isoamyl alcohol (24:1) was added and the samples were mixed thoroughly. Phase separation was carried out by centrifugation at 16,000 g at 4°C. 0.5X Isopropanol and 4 µl of glycogen (5mg/ml) were added to the aqueous phase, and the RNA was pelleted by centrifugation at 16,000 g at 4°C for 30 minutes. The resulting pellet was washed with 70% ethanol, air dried, and

re-suspended in nuclease-free water. All samples of RNAi treated primary cells, metacestode vesicles, and protoscoleces were set up as three independently prepared biological replicates (n = 3).

## Library preparation and sequencing

Library preparation and sequencing were performed at the Wellcome Sanger Institute as recently described [20] using the Illumina TruSeq kit. Briefly, polyadenylated mRNA was purified from total RNA using oligo-dT dynabead selection followed by fragmentation by metal ion hydrolysis. First strand synthesis (using random oligonucleotides) was followed by 2nd strand synthesis with RNaseH and DNApolI. Template DNA fragments were end-repaired with T4 and Klenow DNA polymerases and blunt-ended with T4 polynucleotide kinase. A single 3' adenosine was added to the repaired ends using Klenow exo- and dATP to reduce template concatemerization and adapter dimer formation, and to increase the efficiency of adapter ligation. Adapters containing primer sites for sequencing and index sequences were then ligated. Libraries made with the TruSeq protocol were amplified by PCR to enrich for properly ligated template strands, to generate enough DNA, and to add primers for flow-cell surface annealing, using Kapa HiFi enzyme. AMPure SPRI beads were used to purify amplified templates before pooling based on quantification using an Agilent Bioanalyser chip. Pooled TruSeq libraries were then pooled, and size selected (300–400 bp fragments) using the Caliper. After adaptor ligation, individual libraries made with the Illumina mRNA-seq kit were size selected using the Caliper before PCR amplification followed by AMPure SPRI bead clean up and removal of adaptors with a second Caliper run. Kapa Illumina SYBR Fast qPCR kit was used to quantify the Illumina mRNA-seq libraries before pooling.

Libraries were denatured with 0.1 M sodium hydroxide and diluted to 6 or 8 pM in a hybridization buffer to allow the template strands to hybridize to adapters attached to the flow cell surface. Cluster amplification was performed on the Illumina cBOT using the V4 cluster generation kit following the manufacturer's protocol and then a SYBR Green QC was performed to measure cluster density and determine whether to pass or fail the flow cell for sequencing, followed by linearization, blocking and hybridization of the R1 sequencing primer. The hybridized flow cells were loaded onto the Illumina sequencing platforms for sequencing-by-synthesis (100 cycles) using the V5 SBS sequencing kit then, *in situ*, the linearization, blocking and hybridization step was repeated to regenerate clusters, release the second strand for sequencing and to hybridize the R2 sequencing primer followed by another 100 cycles of sequencing to produce paired end reads. These steps were performed using proprietary reagents according to the manufacturer's recommended protocol. Data was analysed from the Illumina GAIIx or HiSeq sequencing machines using the RTA1.8 analysis pipelines.

## Gene finding, annotation, RNA-Seq mapping, and expression level calculation

Gene finding, annotation, and mapping of sequencing reads has been performed as recently described [20]. Mapping of sequencing reads was performed with Hisat2 v2.0.5 [59] using a maximum intron length of 40,000 base pairs. The number of uniquely mapped reads per transcript was calculated using HTSeqCount v0.7.1 [60] with a minimum quality score of 30. TPM values (Transcripts Per kilobase Million) were calculated for all transcripts. Differential expression was calculated pairwise between datasets using DESeq2 v1.16.1 [61] on statistical computation platform R v3.4.3 [62] with an adjusted p-value cut-off of 0.05. Adjustment for multiple testing was performed using the Benjamini-Hochberg procedure after independent filtering with genefilter v1.58.1 [65] (false discovery rate 0.05). To ascertain quality and correctness of differential expression analysis, fitting of the dispersion curve and outlier detection was assessed by plotting the dispersions (using plotDispEsts function) and the Cook´s distances for each comparison. For improved estimation of actual log2fold changes (FLC), function lfcShrink was used to calculate shrunken maximum a posteriori (MAP) LFCs. For quality control, both unshrunk maximum likelihood estimate (MLE) LFCs and MAP LFCs were visualized by plotting. Unless otherwise indicated, LFC in this work are MAP LFCs.

## Bioinformatic procedures and statistical analyses

BLASTP searches were carried out against the *E. multilocularis* genome (version 5, January 2016) [63] on WormBase ParaSite [64] as well as against the SwissProt database as available at GenomeNet (https://www.genome.jp/). Protein domain structure analysis was carried out using SMART 8.0 (http://smart.embl-heidelberg.de/). Multiple sequence alignments were performed using Clustal Omega (https://www.ebi.ac.uk/Tools/msa/clustalo/) and MEGA 11 software. Statistical analyses were performed using GraphPad Prism (version 9) employing one-way t-test. For statistical analyses on RNAi primary cell cultures, 9 setups of electroporated primary cells (deriving from 3 different primary cell preparations) were analysed and treated as 9 independent biological samples (n = 9). Images to illustrate parasite life cycle and primary cell culture system were drawn by hand using Powerpoint (Microsoft).

## Supporting information

**S1 Fig. Schematic illustration of *E. multilocularis* life-cycle stages.**
(PDF)

**S2 Fig. Schematic illustration of the *E. multilocularis* primary cell cultivation system.**
(PDF)

**S3 Fig. Hyperproliferation of germinative cells in primary cell aggregates after *bcat-1*(RNAi).**
(PDF)

**S4 Fig. Structure and homologies of *Echinococcus ndk* and *ndl* (*ndk*-like) factors.**
(PDF)

**S5 Fig. Structure and homologies of *Echinococcus fst*.**
(PDF)

**S6 Fig. qRT- PCR analysis of gene expression after siPOOL RNAi.**
(PDF)

**S7 Fig. Induction of anterior markers in *bcat-1*(RNAi) culture aggregates.**
(PDF)

**S8 Fig. Structural features of *Echinococcus* MUC2.**
(PDF)

**S9 Fig. Structure and homologies of *Echinococcus* SmadF.**
(PDF)

**S10 Fig. Structure and homologies of *Echinococcus* Notum.**
(PDF)

**S11 Fig. qRT-PCR analysis of gene expression after siPOOL RNAi.**
(PDF)

**S1 Table. Transcriptomic analyses on *bcat-1*(RNAi) cultures, protoscoleces, and metacestode vesicles.**
(XLSX)

**S2 Table. Oligonucleotide primer sequences for WISH and qRT-PCR as well as GenBank accession numbers.**
(XLSX)

**S1 Data. Raw data associated with manuscript figures.**
(XLSX)

## Acknowledgments

The authors are indebted to Raphael Duvoisin and Dirk Radloff for excellent technical assistance.

## Author contributions

**Conceptualization:** Uriel Koziol, Klaus Brehm.

**Data curation:** Michaela Herz, Kilian Rudolf, Markus Spiliotis, Monika Bergmann, Nancy Holroyd, Uriel Koziol, Matt Berriman, Klaus Brehm.

**Formal analysis:** Ruth Herrmann, Michaela Herz, Uriel Koziol, Matt Berriman, Klaus Brehm.

**Funding acquisition:** Matt Berriman, Klaus Brehm.

**Investigation:** Ruth Herrmann, Michaela Herz, Kilian Rudolf, Akito Koike, Markus Spiliotis, Monika Bergmann, Nancy Holroyd, Uriel Koziol, Klaus Brehm.

**Methodology:** Ruth Herrmann, Markus Spiliotis, Monika Bergmann, Nancy Holroyd, Uriel Koziol, Matt Berriman.

**Project administration:** Nancy Holroyd, Matt Berriman, Klaus Brehm.

**Resources:** Nancy Holroyd.

**Supervision:** Matt Berriman, Klaus Brehm.

**Validation:** Ruth Herrmann, Michaela Herz, Akito Koike, Markus Spiliotis, Nancy Holroyd, Uriel Koziol, Matt Berriman, Klaus Brehm.

**Visualization:** Michaela Herz, Kilian Rudolf, Uriel Koziol, Klaus Brehm.

**Writing – original draft:** Klaus Brehm.

**Writing – review & editing:** Ruth Herrmann, Uriel Koziol, Matt Berriman, Klaus Brehm.

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
