## [Decision Letter · Decision Letter 0]

3 Dec 2025

Canonical WNT signalling governs Echinococcus metacestode development

PLOS Pathogens

Dear Dr. Brehm,

Thank you for submitting your manuscript to PLOS Pathogens. After careful consideration, we feel that it has merit but does not fully meet PLOS Pathogens's publication criteria as it currently stands. Therefore, we invite you to submit a revised version of the manuscript that addresses the points raised during the review process.

We look forward to receiving your revised manuscript.

Kind regards,

Richard J. Martin, BVSc, PhD, DSc, DipECVPT, FRCVS

Guest Editor

PLOS Pathogens

Edward Mitre

Section Editor

Editor-in-Chief

PLOS Pathogens

orcid.org/0000-0003-2946-9497

Editor-in-Chief

PLOS Pathogens

orcid.org/0000-0002-7699-2064

**Additional Editor Comments :**

The manuscript has been reviewed by 3 expert referees and an editor who see merit in your study of WNT signalling in Echinococcus metacestode development. Each of the referees point out aspects of the report that need to be addressed before publication can be considered further. Accordingly, you are advised to review and consider carefully each of their comments.

**Journal Requirements:**

1) We do not publish any copyright or trademark symbols that usually accompany proprietary names, eg ©,  ®, or TM  (e.g. next to drug or reagent names). Therefore please remove all instances of trademark/copyright symbols throughout the text, including:

- ® on page: 29.

2) Please amend your detailed Financial Disclosure statement. This is published with the article. It must therefore be completed in full sentences and contain the exact wording you wish to be published.

3) Please ensure that the funders and grant numbers match between the Financial Disclosure field and the Funding Information tab in your submission form. Note that the funders must be provided in the same order in both places as well.

**Reviewers' Comments:**

Reviewer's Responses to Questions

**Part I - Summary**

Reviewer #1: The manuscript by Herrmann et al builds upon that of Jarero et al (2024) and Koziol et al (2016) to demonstrate the importance of β-catenin (bcat-1) and, thereby, canonical Wnt signalling in Echinococcus multilocularis metacestode development. The authors utilise RNAi to silence bcat-1 in primary cell cultures in vitro, resulting in impaired vesicle formation and increased stem cell proliferation. Transcriptomic analyses revealed an anteriorization of gene expression in bcat-silenced groups and this was corroborated by WISH. Phenotypic effects following treatment with the Wnt antagonist pyrvinium pamoate are also reported in protoscoleces and notably, this proves lethal.

In some cases, the interpretation of results could be supported more robustly by relevant literature and the authors should explain the rationale behind performing experiments (intended to complement one another) on different developmental stages. Overall, the data corroborate what has been reported in related flatworm species following β-catenin knockdown and provide a valuable insight into the developmental biology of a cestode of significant medical importance. The phenotypes observed following bcat knockdown and treatment with PP suggest that both have potential in the development of novel therapeutic strategies.

Reviewer #2: In the manuscript entitled “Canonical WNT signalling governs Echinococcus metacestode development” Herrmann, Schiegl et al. study the effect of inhibition of the Wnt signalling during Echinococcus metacestode development. The authors inhibit beta-catenin by RNAi in primary cell cultures that produce metacestode vesicles and observe the upregulation of genes related with anterior specification and the downregulation of genes related with posterior. They also observe defects in the muscular cells and in the proliferation of stem cells. Treatment of the same cultures with pyrvinium pamoate, a canonical Wnt signal inhibitor, produce defects in the maintenance of the cultures and stem cell proliferation. The authors conclude that the Wnt/beta-catenin signal directs Echinococcus body axis formation and the posteriorization events that are observed during the growth of the metacestode within the host.

The study is interesting, since it could help to identify new drugs to treat the infection. According to the previous publications, it is possible that the posteriorizing effect of the Wnt signaling pathway could be controlling the development of the metacestode, and thus it’s inhibition could be considered as a strategy to cure infections. It is also important the achievement of RNAi inhibition of a target gene in this species. To obtain functional data of the Wnt signaling pathway in Echinococcus development is difficult and merits a recognition.

The weakness of the manuscript is the lack of a proper visual introduction of the different developmental stages of Echinococcus, which make sit very difficult to follow the experiments. Furthermore, the images and not well presented in terms of magnifications, labeling and indication of the structures analyzed in each one.

A second weakness is the experiment related with the drug treatment, since the results seem unspecific and their relationship with the Wnt signaling pathway are not demonstrated.

To conclude, the study has interest for the readers but should be improved in terms of experiments and presentation of the data.

Reviewer #3: The research group is recognized for pioneering Echinococcus multilocularis as a laboratory model. They established long‑term culture systems of metacestode vesicles and primary cell cultures derived from the germinal layer, enabling direct genetic and pharmacological studies. They also refined RNA interference (RNAi) protocols for proliferative parasite stages, demonstrating reproducible gene silencing, and integrated genome‑wide transcriptomics, qRT‑PCR, and whole‑mount in situ hybridization (WMISH) into a robust validation framework. Importantly, they determined that canonical WNT signaling is conserved and functional in parasitic flatworms, albeit represented by a reduced set of WNT paralogs, and described the Echinococcus metacestode larva as a broadly posteriorized tissue.

Building on this foundation, the present study interrogates the role of β‑catenin (bcat‑1), the central effector of canonical WNT signaling, in metacestode formation and axial identity. RNAi knockdown of bcat‑1 was performed in primary parasite cell cultures (isolate Ingrid) that recapitulate metacestode development in vitro. Two strategies were applied—conventional siRNAs and the stringent siPOOL system, which minimizes off‑target effects by distributing silencing activity across complex pools of low‑concentration duplexes. Both approaches achieved stable reduction of bcat‑1 expression to <50% of control levels for at least 14 days.

The hallmark outcome was the reproducible “red‑dot phenotype”: cultures arrested at the early cavity stage, retaining phenol‑red staining, with stem‑cell hyperproliferation (~32% S‑phase vs. ~15% in controls) and disrupted muscle‑fiber organization. This phenotype was robust across both RNAi strategies and represents the visible manifestation of developmental arrest and generalized anteriorization when the posteriorizing cWNT pathway is disrupted.

Genome‑wide transcriptomics confirmed broad anteriorization: anterior positional control genes (sfrp, fz10, ndk, notum, fst) were upregulated and localized to the presumptive anterior pole, while posterior/metacestode‑specific genes (muc‑1, TNFR, Antigen B, post2b) were significantly downregulated. These transcriptomic shifts were validated independently by qRT‑PCR and WMISH, underscoring the robustness of the findings.

Pharmacological inhibition provided independent confirmation. Pyrvinium pamoate (PP), a CK1α activator that stabilizes the β‑catenin destruction complex, induced concentration‑dependent posterior loss (“tailless” phenotype) in protoscoleces, compromised vesicle integrity at 2 µM, and suppressed stem‑cell proliferation at concentrations as low as 20 nM. PP was particularly appropriate compared to upstream inhibitors like IWP‑3, because it acts downstream, ensuring pathway inhibition even when multiple WNT paralogs are present.

The methodological strength of the study is striking: RNAi experiments were performed with three independently prepared biological replicates (N=3), statistical analysis was based on nine independent samples, and PP assays were conducted with three independent cultures (N=3), each with three technical replicates (n=3). Data availability was explicitly declared, facilitating verification by the scientific community.

The convergence of dual RNAi strategies, genome‑wide transcriptomics, qRT‑PCR, WMISH, and pharmacological inhibition demonstrates with exceptional rigor that canonical WNT signaling, mediated by bcat‑1, is indispensable for posteriorization and metacestode development in E. multilocularis. β‑catenin functions as a non‑redundant effector in body‑axis formation and larval proliferation.

This study provides a compelling and methodologically rigorous demonstration that canonical WNT signaling is a central posteriorizing force in Echinococcus multilocularis. The careful design, replication, and multi‑angle validation framework make its conclusions solid and persuasive, while also highlighting CK1α and the β‑catenin destruction complex as promising molecular targets for anti‑AE chemotherapy.

Because RNAi directly targets β‑catenin rather than WNT ligands or receptors, and because PP acts downstream in ways that may influence β‑catenin pools beyond nuclear/TCF signaling, certain phenotypes—such as muscle‑fiber disorganization—could partly reflect β‑catenin’s WNT‑independent functions. In addition to serving as a transcriptional effector, β‑catenin contributes to cell adhesion at adherens junctions and to cytoskeletal organization. Even so, the study convincingly demonstrates that canonical WNT signaling is the central regulator of posteriorization in Echinococcus multilocularis, while acknowledging that β‑catenin and its regulator CK1α may also affect stem‑cell proliferation through WNT‑independent mechanisms, including crosstalk with the Hippo pathway. By situating their findings within this broader signaling context, the authors reinforce the conserved role of WNT in axis formation and highlight promising directions for Evo‑Devo and parasitology, where Hippo–WNT interactions and neuropeptide regulation may emerge as novel dimensions of parasite development and therapeutic targeting.

One of the most impressive aspects of this work is how much more it has been able to extract from the already well‑established Echinococcus multilocularis model. The parasite has long been recognized as a tractable system for studying metacestode biology, but the authors’ multilayered strategy—combining dual RNAi approaches, genome‑wide transcriptomics, qRT‑PCR, WMISH, and pharmacological inhibition—pushes the model far beyond its traditional use. This integration not only confirms the centrality of canonical WNT signaling in posteriorization and larval development, but also demonstrates the power of Echinococcus as a platform for functional genomics in cestodes. The convergence of independent lines of evidence makes the conclusions unusually robust and sets a new benchmark for how deeply developmental pathways can be interrogated in parasitic flatworms.

**Part II – Major Issues: Key Experiments Required for Acceptance**

Please use this section to detail the key new experiments or modifications of existing experiments that should be absolutely required to validate study conclusions.required to validate study conclusions.

Reviewer #1: NA

Reviewer #2: Major issues:

-The development of the different forms of Echinococcus is explained in the introduction. However, to understand the results obtained it would be necessary to show a scheme indicating the different stages analyzed during this work. In addition, it is necessary that the authors include a scheme of the corresponding stages and forms that are analyzed in each Figure. They should also indicate the different cell types or tissues, and the orientation of each analyzed form.

For instance: Lines 233-235 “Because bcat-1(RNAi) was expected to affect body-axis formation, we also generated datasets for the protoscolex (PS), which exhibits an A–P axis [5], and for metacestode vesicles devoid of brood capsules, which represent posteriorized tissue [5].” Sentences like that are not understandable without a previous definition and explanation of each stage and tissue.

It should be clarified what is exactly analyzed in each image of each figure.

-The authors should include an explanation of what are Echinococcus primary cell cultures. Which kind of cells can be found in the culture? How metacestode vesicles are organized in relation to a in vivo metacestode?

-The experiment with the pyrvinium pamoateare is not informative:

->The Tailles phenotype in protoscolex does not indicate that the drug is inhibiting the Wnt pathway. Markers of anterior and posterior identity, or targets of the Wnt signaling, should be used. Also, the genes found differentially expressed in the previous strategy could be used as markers to demonstrate that the drug has some effect at the level of the Wnt signalling.

->Lines 481-483, The tailless phenotype was described in planarians, as a result of Wnt/beta-catenin inhibition. In these phenotypes wnt1 is decreased or not expressed. The authors have no evidences to argue that “additional wnt1-expressing cells appeared at the wound site, indicating an attempt to compensate for the loss of the original posterior pole.”

->The disintegration of the vesicles after PP treatment is not a specific phenotype.

->The drug has different effect on proliferating cells with the drug than the beta-catenin RNAi in the cultures. This result does not support any specific action of the drug in inhibiting the Wnt signaling in this context.

These results suggest that the effect of the drug is not specific, at least at these concentrations. This part of the manuscript should be improved with the use of specific markers and the demonstration of a Wnt-specific phenotype, or completely removed.

Reviewer #3: None. The study is very solid and convincing; there are no validations that should be absolutely required.

**Part III – Minor Issues: Editorial and Data Presentation Modifications**

Reviewer #1: l. 46 – In situ should be italicised here and elsewhere in the manuscript.

1. 137 - In vitro should be italicised here and elsewhere in the manuscript.

Fig. 1, B and D – Please display/state level of significance.

Fig. 1, E legend – No need to state “(red)” as does not apply to graph.

l. 205 – Which group was used for staining, siRNAs or siPOOLs?

l. 203-204 – Vesicle formation in siPOOL control group is reduced compared to siRNA control group – authors should consider possibility of off-target/toxic effects? Could such effects have contributed to the observation that mature vesicle formation was “almost completely abolished” in the siPOOL bcat group and not in the bcat siRNA group? On this basis, the value of including siPOOL data is questionable.

l. 210 - 215 – The dysregulation of development and muscle organisation has also been reported following bcat knockdown in H. diminuta (Nanista et al., 2025) and F. hepatica (Armstrong et al., 2025) – it would seem most appropriate to reference these here.

l. 219. Again, siRNAs or siPOOLs? Please make this clear.

l. 230 – “These data indicate that the increased number of EdU⁺ cells reflect a general over-representation of stem cells in culture, not merely a short-term increased proliferation rate of otherwise constant numbers of stem cells.” – upregulation of 35.7% of stem cell-associated transcripts does not adequately support this statement. It would seem important to consider including additional evidence, e.g. EdU/BrdU pulse-chase in control vs bcat RNAi, or remove statement.

l. 374 -379 – An image showing lack of expression of fz10 expression in a fully developed protoscolex, at least in supplementary info would be helpful.

Fig 4 – If s = suckers, please state this in legend.

l. 433. Co-localisation with a muscle cell marker would be beneficial to support this statement.

l. 477 – 2 µM - change to 2000 nM to be consistent with figure.

l. 479 - 486 – Were these changes observed at day 2, 4 or 7?

Fig 6 – How many days post treatment are the metacestodes in the images? Include this info in legend.

Fig 6, I – Looks like there should be significance at d4 and d7 for 2000 nM? Displaying the lower limit of error bars here (and in all other graphs presented) would be helpful.

l. 528 – until *the end of *the experiment.

Fig 7, I – Significance at 20 nM, but not 200 nM – please state possible explanation.

l. 618 - 619 – The generation of tegumental and hydatid-fluid components is multifaceted and disruption of the tegument following bcat RNAi was not evidenced in images – please adjust or remove statement to reduce speculation.

l. 635 – While the data are suggestive of cWNT signalling dysregulation, PP is also believed to inhibit glucose uptake and mitochondrial fumarate reductase which may produce a similar phenotype – this should be acknowledged.

l. 656-669 – This section of text is highly speculative – please reduce.

l. 680 – Using primary cell cultures for RNAi and protoscoloces for PP experiments diminishes the value of phenotypic comparisons – please explain rationale.

l. 687, 690, 700, 750, 766 – “essentially as previously described” – any deviations from the protocol referenced should be clearly stated.

Reviewer #2: These are issues which are not really minor because they should be addressed before publication, but they are not experiments:

-Figure 1 L-M, magnifications of EdU stained cells should be showed, to demonstrate the signal in the nucleus.

-Why is the “red-dot phenotype” obtained after beta-catenin inhibition? Showing magnifications and detailed explanations would help.

-In Figure 1, F-M, add labels to know what is control and RNAi. The same applies for figures 4-5-6-7. Add the concentration of the drug used, and the condition in each image. In Figure 5 add EdU in red in the images.

-In Figure3, the magnification in Controls and RNAis of the same gene should be the same to be comparable.

-Lines 210-212 ‘Moreover, muscle fibres in bcat-1(RNAi) aggregates were disorganized and only rarely accumulated around central cavities, a pattern typically observed during vesicle formation in this system (Fig 1J-M). The disorganization of muscle fibers is not evident. A magnification or an image of muscle fibers in the metacestode would help.

-In Figure 1 J-K it seems that there are more neural cells in the RNAi than in the control. Authors could quantify it because it is very obvious in this image. If this is the case, it would fit with the anteriorized phenotype.

- What is the relationship between the transcriptomic switch towards anterior genes and the phenotype of the vesicles (red vesicles, disorganized muscle…)? The authors should discuss it.

-After beta-catenin inhibition the authors find an increase in proliferation in the cultures (It should it be J-K instead of J-M). Is it an expected result? Discuss it.

-Lines 356-358. “Taken together, these analyses show that multiple hallmark components crucial for metacestode vesicle development and physiology are profoundly suppressed in primary cell cultures upon bcat-1 knockdown.” To what extent the effects seen in the primary cell cultures are applicable to an in vivo metacestode? This is an important issue that should be discussed. As stated, the authors should clarify what are the similarities and differences between the cultures and the in vivo structures to be able to discuss this issue.

Reviewer #3: While the manuscript is methodologically rigorous and compelling and the study convincingly shows dose‑dependent effects of PP on Echinococcus protoscoleces and metacestode vesicles, the rationale for the chosen concentration range (20, 200 and 2000 nM), exposure schedules, and replenishment protocols is not fully detailed. I would appreciate more detail regarding these experiments. The study convincingly demonstrates dose‑dependent effects of PP but the basis for selecting the PP design is not fully explained. Clarifying whether these regimes were chosen based on undisclosed pilot studies in Echinococcus, extrapolated from related helminths (for S. mansoni, a single 1000-nM dose was used in a compound screening (DOI: 10.1371/journal.pntd.0000478), and in the cited Fasciola study (ref. 20), PP was used at 100 nM with mention of other “over 100 nM doses”) or derived from pharmacological precedent in other systems would strengthen the transparency of the experimental design. Given that other cestodes and trematodes have a significantly more limited experimental toolbox, fuller reporting of how PP was applied in Echinococcus is essential to enable adaptation of these methods across species. Clearer explanation of the design choices would make the work even more valuable as a methodological reference for the wider helminth research community.

A minimal aspect, at fine-grain level, is that the quantitative RT-PCR used only one gene (elp, EmuJ_000485800) for normalization, instead of at least two or three, according to MIQE although it is backed by consistent use by the group. The use of elp (EmuJ_000485800) as a reference gene is justified, as it has been validated by the research group in Echinococcus larval stages and primary cell cultures, and shown to be unaffected by electroporation (Herz et al. 2024; Pérez et al. 2019, 2022). However, MIQE guidelines recommend multiple reference genes for normalization. I suggest the authors explicitly cite their prior validation of elp and briefly discuss MIQE compliance, so that readers can better appreciate the robustness of their qRT‑PCR normalization strategy and adapt it to other cestodes with limited validated reference genes.

Editorially:

Line 177/178, Line 185: Homogenize decimals. Periods are used throughout the text and supporting tables, but some commas should be corrected: p < 0,0001

Lines 493, 517, 522, 525, 527 : The legends of Figures 6 and 7, which describe the effects of PP on protoscolices and metacestode vesicles, use the abbreviation "µM" or "uM" for 20, 200, and 2000 nM (nanomolar), described in the Methods section.

Line 511 refers to the 2000 nM concentration as 2µM, which is correct, but for homogeneity, it could be better all expressed as nM.

Line 524: The scale bar should be 50 µM, not 50 uM.

Line 848 (n = 3) instead of (N = 3), like the others.

S4 figure, italicize gene names fz10, ndk, fst, and sfrp. correct "indicaterd" in the legend.

PLOS authors have the option to publish the peer review history of their article (what does this mean? ). If published, this will include your full peer review and any attached files.). If published, this will include your full peer review and any attached files.

**Do you want your identity to be public for this peer review?** For information about this choice, including consent withdrawal, please see our For information about this choice, including consent withdrawal, please see our Privacy Policy ..

Reviewer #1: No

Reviewer #2: No

Reviewer #3: No

**Figure resubmission:**

**Reproducibility:**



---

## [Editor Report · Decision Letter 1]

28 Feb 2026

Dear Prof. Brehm,

We are pleased to inform you that your manuscript 'Canonical WNT signalling governs Echinococcus metacestode development' has been provisionally accepted for publication in PLOS Pathogens.

Best regards,

Richard J. Martin, BVSc, PhD, DSc, DipECVPT, FRCVS

Guest Editor

PLOS Pathogens

Edward Mitre

Section Editor

PLOS Pathogens

Sumita Bhaduri-McIntosh

Editor-in-Chief

PLOS Pathogens

orcid.org/0000-0003-2946-9497

Michael Malim

Editor-in-Chief

PLOS Pathogens

orcid.org/0000-0002-7699-2064

The reviewers see merit in this manuscript that demonstrates the importance of β-catenin (bcat-1) and canonical Wnt signaling in Echinococcus multilocularis metacestode development. The phenotypes observed following bcat knockdown have potential for the development of novel therapeutic strategies.

The reviewers had indicated that more robust and relevant literature to explain the reasons for the experiments performed would be beneficial and there was a lack of a proper visual introduction of the different developmental stages of Echinococcus, which made it difficult to follow the experiments. There were concerns about the specificity of pyrvinium pamoate. In addition, specific phrasing and editorial suggestions for improvement were suggested.

The authors have responded appropriately to the critiques of the reviewers and the authors have included additional supplementary figures and references to make the task of following the experiments easier and have included additional relevant literature. They have removed the entire pyrvinium pamoate section from the revised manuscript.
---

## [Editor Report · Acceptance letter]

Dear Prof. Brehm,

We are delighted to inform you that your manuscript, "Canonical WNT signalling governs Echinococcus metacestode development," has been formally accepted for publication in PLOS Pathogens.

Best regards,

Sumita Bhaduri-McIntosh

Editor-in-Chief

PLOS Pathogens

orcid.org/0000-0003-2946-9497

Michael Malim

Editor-in-Chief

PLOS Pathogens

orcid.org/0000-0002-7699-2064